# GRAPH NEURAL NETOWRK POOLING BY EDGE CUT

## ABSTRACT

Graph neural networks (GNNs) are very efficient at solving several tasks in graphs such as node classification or graph classification. They come from an adaptation of convolutional neural networks on images to graph structured data. These models are very effective at finding patterns in images that can discriminate images from each others. Another aspect leading to their success is their ability to uncover hierarchical structures. This comes from the pooling operation that produces different versions of the input image at different scales. The same way, we want to identify patterns at different scales in graphs in order to improve the classification accuracy. Compared to the case of images, it is not trivial to develop a pooling layer on graphs. This is mainly due to the fact that in graphs nodes are not ordered and have irregular neighborhoods. To aleviate this issue, we propose a pooling layer based on edge cuts in graphs. This pooling layer works by computing edge scores that correspond to the importance of edges in the process of information propagation of the GNN. Moreover, we define a regularization function that aims at producing edge scores that minimize the minCUT problem. Finally, through extensive experiments we show that this architecture can compete with state-of-the-art methods.

## 1 INTRODUCTION

Convolution neural networks (LeCun et al., 1995) have been proven to be very efficient at learning meaningful patterns for many articificial intelligence tasks. They convey the ability to learn hierarchical informations in data with Euclidean grid-like structures such as images and textual data. Convolutional Neural Networks (CNNs) have rapidly become state-of-the-art methods in the fields of computer vision (Russakovsky et al., 2015) and natural language processing (Devlin et al., 2018).

However in many scientific fields, studied data have an underlying graph or manifold structure such as communication networks (whether social or technical) or knowledge graphs. Recently there have been many attempts to extend convolution to such non-Euclidean structured data (Hammond et al., 2011; Kipf & Welling, 2016; Defferrard et al., 2016). In these new approaches, the authors propose to compute node embeddings in a semi-supervised fashion in order to perform node classification. Those node embeddings can also be used for link prediction by computing distances between each node of the graph (Hammond et al., 2011; Kipf & Welling, 2016).

An image can be seen as a special case of graph that lies on a 2D grid and where nodes are pixels and edges are weighted according to the difference of intensity and to the distance between two pixels (Zhang et al., 2015; Achanta & Susstrunk, 2017; Van den Bergh et al., 2012; Stutz et al., 2018). In the emerging field of graph analysis based on convolutions and deep neural networks, it is appealing to try to apply models that worked best in the field of computer vision. In this effort, several ways to perform convolutions in graphs have been proposed (Hammond et al., 2011; Kipf & Welling, 2016; Defferrard et al., 2016; Gilmer et al., 2017; Veličković et al., 2017; Xu et al., 2018; Battaglia et al., 2016; Kearnes et al., 2016). Moreover, when dealing with image classification, pooling is an important step (Gao & Ji, 2019; Ying et al., 2018; Defferrard et al., 2016; Diehl, 2019). It allows us to extract hierarchical features in images in order to make the classification more accuracte. While it is easy to apply coarsening to an image, it isn't obvious how to coarsen a graph since nodes in graphs are not ordered like pixels in images. In this work we present a novel pooling layer based on edge scoring and related to the minCUT problem.

The main contributions of this work are summarized below:

1. **Learned pooling layer.** A differentiable pooling layer that learns how to aggregate nodes in clusters to produce a pooled graph of reduced size.

2. **A novel approach based on edge cuts.** We develop a novel pooling layer. Most coarsening strategies are based on nodes, either by finding clusters or by deleting nodes that carry less information of the graph structure. In our approach, we focus on edges to uncover communities of topologically close nodes in graphs.

3. **The definition of a regularization that aims at approximating the problem of minCUT.** We regularize our problem by a term that corresponds to the problem of Ncut in order to learn edge scores and clusters that are consistent with the topology of the graph. We show that by computing an edge score matrix, we can easily compute this regularization term.

4. **Experimental results.** Our method achieves state-of-the-art results on benchmark datasets. We compare it with kernel methods and state-of-the-art message passing algorithms that use pooling layers as aggregation processes.

## 2 RELATED WORK

Recently there has been a rich line of research, inspired by deep models in images, that aims at redefining neural networks in graphs and in particular convolutional neural networks (Defferrard et al., 2016; Kipf & Welling, 2016; Veličković et al., 2017; Hamilton et al., 2017; Bronstein et al., 2017; Bruna et al., 2013; Scarselli et al., 2009). Those convolutions can be viewed as message passing algorithms that are composed of two phases Gilmer et al. (2017). They find their success in their ability to uncover meaningful patterns in graphs by propagating information from nodes to their neighbors. Moreover, many works on graph neural networks also focus on redefining pooling in graphs. The pooling operation allows us to obtain different versions of the input graph at different scales. In graphs, the pooling step isn't trivial because of the nature the data. Nodes can have different numbers of neighbors and graphs can have different sizes. To cope with these issues, different pooling strategies have been proposed:

- **Top-k**: Like Gao & Ji (2019), the objective is to score nodes according to their importance in the graph and then to keep only nodes with the top-k scores. By removing nodes we can remove important connections in the graph and produce disconnected graphs. A step to increase connectivity is necessary. This is done by adding edges at 2-hops from the input graph.

- **Cluster identification**: This is usually done by projecting node features on a learned weight to obtain an assignment matrix. Nodes that have close embeddings are projected on the same cluster. After having obtained the assignment matrix, super nodes at the coarsened level can be computed by aggregating all nodes that belong to the same cluster (Ying et al., 2018).

- **Edge based pooling**: An edge contraction pooling layer has recently been proposed by Diehl (2019). They compute edge scores in order to successively contract pairs of nodes, which means that they successively merge pairs of nodes that are linked by edges of the highest scores.

- **Deterministic coarsening strategies**: Finally, a way to perform pooling in graphs can simply be to apply a deterministic clustering algorithm in order to identify clusters of nodes that will represent super nodes in the coarsened level (Defferrard et al., 2016; Ying et al., 2018). The main drawback of it is that the strategy isn't learned and thus may not be suited to the graph classification task.

In this work we define a new pooling layer that is based on edge cuts. Like Diehl (2019) we focus our pooling method on edges instead of nodes. In their work, Diehl (2019) calculate scores on edges to perform contraction pooling. This means that at each pooling step, they merge pairs of nodes that are associated with the highest edge scores, without merging nodes that were already involved in a contracted edge. This method results in pooled graphs of size divided by 2 compared to the input graph.
The main similarity with our work is that we compute edge scores to characterize edge importance inspired by Graph Attention Transform (Veličković et al., 2017). There are several differences with

the pooling layer that we propose in this work. We want our pooling layer not to be constrained by a number of communities or by a predefined size of pooled graph. Moreover, our pooling layer works by edge cuts and the goal is to remove edges that minimize the minCUT problem (Stoer & Wagner, 1997). Once edges are cut, the graph is no longer connected and is composed of several connected components. These connected components correspond to super nodes in the coarsened level. In this work, we will first introduce the pooling architecture based on edge scoring in section 3.1. We will then relate this pooling layer to the minCUT problem in section 3.2. We will finally compare this pooling layer to state-of-the-art methods on benchmark datasets on a graph classification task and a node classification task in section 4.

# 3 POOLING ARCHITECTURE

When designing a pooling layer, most algorithms need a number of classes for the pooling layer that is usually set as a hyperparameter. This is very restrictive especially when working on graphs of different sizes. Indeed, the pooling layer should cluster nodes according to the topology of the graph without being constrained by a number of classes. In this section we present our pooling layer that is based on edge cutting and that does not necessitate any *a priori* on the number of classes that needs to be found.

## 3.1 GNNs

Let $G = (V, E, X)$ be a graph composed of a set of nodes $V$, a set of edges $E$ and a node feature matrix $X \in \mathbb{R}^{n \times f_0}$ where $f_0$ is the dimensionality of node features. We denote by $A$ the adjacency matrix.

**Graph neural networks.** We build our work upon graph neural networks (GNNs). There are several architectures of graph neural networks that have been proposed by Defferrard et al. (2016); Kipf & Welling (2016); Veličković et al. (2017) or Bruna & Li (2017). Those graph neural network models are all based on propagation mechanisms of node features that follow a general neural message passing architecture (Ying et al., 2018; Gilmer et al., 2017):

$$Z^{(l+1)} = MP(A, Z^{(l)}; W^{(l)}) \tag{1}$$

where $Z^{(l)} \in \mathbb{R}^{n \times f_l}$ are node embeddings computed after $l$ steps of $MP$, $Z^{(0)} = X$, and $MP$ is the message propagation function, which depends on the adjacency matrix. $W^{(l)}$ is a trainable weight matrix that depends on layer $l$ and $f_l$ is the dimensionality of node embeddings.

The pooling layer that we introduce next can be used with any neural message passing algorithm that follows the propagation rule 1. In all the following of our work we denote by $MP$ the algorithm. For the experiments, we consider the Graph Convolutional Network (GCN) defined by (Kipf & Welling, 2016). This model is based on an approximation of convolutions on graphs defined by (Defferrard et al., 2016) and that use spectral decompositions of the Laplacian. It is very popular because it is very efficient computationally and obtains state-of-the-art results on benchmark datasets. This layer propagates node features to 1-hop neighbors. Its propagation rule is the following:

$$Z^{(l+1)} = MP(A, Z^{(l)}; W^{(l)}) = GCN(A, Z^{(l)}) = \rho(\tilde{D}^{-1/2} \tilde{A} \tilde{D}^{-1/2} Z^{(l)} W^{(l)}) \tag{2}$$

Where $\rho$ is a non-linear function (a $ReLU$ in our case), $\tilde{A} = A + I_n$ is the adjacency matrix with added self-loops and $\tilde{D}_{ii} = \sum_j \tilde{A}_{ij}$ is the degree diagonal matrix associated with adjacency matrix $\tilde{A}$.

**Scoring edges.** After layer $l$, each node $i$ in the graph has an embedding $Z_i^{(l)}$. To simplify notations, we consider all matrices to be associated to layer $l$ and we do not keep the exponant $l$. For example, we write feature of node $i$ at layer $l$, $Z_i$ and its dimensionality is denoted by $f$. Based on these embeddings, we develop a scoring function that characterizes the importance of each edge of the graph. The input of our scoring algorithm is a set of node features, $\{Z_1, ..., Z_n\} \in \mathbb{R}^{n \times f}$. The scoring function produces a matrix $S \in \mathbb{R}^{n \times n}$ associated with layer $l$, $S_{ij} = \mathbf{1}_{(i,j) \in E} * s_{ij}$ where $s_{ij}$ is the score of edge $(i, j)$.

In order to compute the score of each edge of the graph, we apply a shared linear transformation, parametrized by a weight matrix $W_{pool} \in \mathbb{R}^{f \times d}$, to each node of the graph, $d$ being the output size of the linear transformation. We then perform self-attention on nodes, as used in the Graph Attention Network (GAT) (Veličković et al., 2017), by applying a shared weight $a : \mathbb{R} \times \mathbb{R} \to \mathbb{R}$ to obtain a score on edge $(i, j) \in E$:

$$s_{ij} = \sigma(a[W_{pool}Z_i || W_{pool}Z_j]) \tag{3}$$

Where $\sigma$ is the sigmoid function, $W_{pool}$ and $a$ are trainable matrices associated with layer $l$ and $[W_{pool}Z_i || W_{pool}Z_j] \in \mathbb{R}^{2d}$ is a vector that is the concatenation of $W_{pool}Z_i$ and $W_{pool}Z_j$. Let's note that this scoring function isn't symmetric and depends on the order of nodes. We can symmetrize this function by computing

$$s_{ij} = \tfrac{1}{2} \left( \sigma(a[W_{pool}Z_i || W_{pool}Z_j]) + \sigma(a[W_{pool}Z_j || W_{pool}Z_i]) \right)$$

By applying the sigmoid function to the attention mechanism we compute an importance of edges. The goal is to obtain a distribution on edges for whose nodes that are close topologically have an edge which value is close to 1. In the opposite case, we would like an edge to have a weight close to 0 if it links two nodes that do not lie in the same community. By doing so we would like to solve the minimum cut problem in graphs.

After having computed the edge score matrix, we keep a ratio $r$ of edges that correspond to edges with the $r\%$ higher scores. We obtain a threshold $s_{threshold}$ that corresponds to the $r^{th}$ percentile of the distribution of edge scores. This way, we cut edges which scores are close to 0 in the graph. Edges with the smallest scores represent edges that link nodes that aren't in the same community and thus by cutting those edges, we separate the graph into several clusters. We denote by $S_{cut}$ the score matrix with values under $s_{threshold}$ truncated to 0. Each row is renormalized by the number of positive components. This renormalization is useful in the following to compute node features in the coarsened level.

$$\forall (i, j) \in V^2, S_{cut\,ij} = \frac{1}{\sum\limits_{j \in \mathcal{N}(i)} \mathbf{1}_{s_{ij} \geq s_{threshold}}} s_{ij} \mathbf{1}_{s_{ij} \geq s_{threshold}}$$

We then extract the connected components of the new graph with cutted edges. Those connected components represent super nodes in the pooled graph. We obtain a cluster assignment matrix $C \in \mathbb{R}^{n \times c}$, $c$ being a free parameter that isn't fixed and that can vary during the training of the algorithm. After layer $l$, the pooled adjacency matrix and the pooled feature matrix are thus:

$$A^{(l+1)} = A^{(l)}_{pool} = C^{(l)^T} A^{(l)} C^{(l)}$$
$$Z^{(l+1)} = Z^{(l)}_{pool} = C^{(l)^T} S^{(l)}_{cut} Z^{(l)}$$

**Remark.** The multiplication by $S_{cut}Z$ makes the weights $W_{pool}$ and $a$ trainable by back-propagation. Otherwise it wouldn't be the case because the function that outputs the matrix $C$ by finding connected components from matrix $S_{cut}$ is not differentiable.
Moreover, this multiplication weights the importance of each node feature in the super node of the coarsened level. In order to compute the feature $Z_k$ of cluster (or super node) $k$, we compute a node importance score $s_{cut\,i}$ at layer $l$ for each node $i$ of the graph:

$$s_{cut\,i} = \frac{1}{\sum\limits_{j \in \mathcal{N}(i)} \mathbf{1}_{s_{cut_{ij}^{(l)}} > 0}} \sum_{j \in \mathcal{N}(i)} s_{cut\,ij}$$

The feature $Z_k$ of cluster $k$ is then a weighted mean of the features of nodes that belong to cluster $k$:

$$Z_k = \sum_{i \in k} s_{cut\,i} Z_i$$

Moreover, for edge scores to be consistent with the minCUT algorithm, we add a regularization term that we define in the next section.

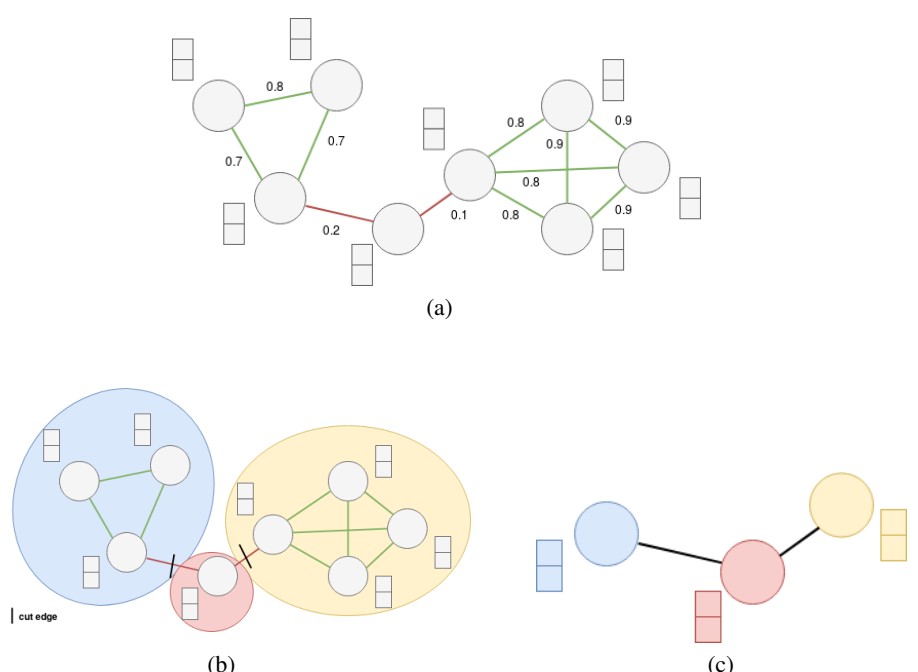

Figure 1: In figure 1a, scores are computed and edges with the smallest scores (in red) are cut. After having cut edges, we identify connected components in figure 1b (black lines on edges mean that those edges have been removed). We finally merge nodes that belong to the same cluster and reconstruct the edges between clusters as shown in figure 1c.

## 3.2 MIN CUTS

**Graph Cuts.** In community detection , the min cut problem aims at finding the separation, between disjoint subsets, that minimizes the cut. For two disjoint subsets $V_1, V_2 \subset V$ we define

$$cut(V_1, V_2) = \sum_{i \in V_1, j \in V_2} A_{ij}$$

Given a similarity graph with adjacency matrix $A$, the simplest and most direct way to construct a partition is to solve the mincut problem. This consists of choosing the partition $V_1, ..., V_c$ of $V$ which minimizes

$$cut(V_1, ..., V_c) = \sum_{k=1}^{c} cut(V_k, \bar{V}_k)$$

When solving this problem, the solution often boils down to separating one indivual vertex from the rest of the graph. In order to circumvent this problem, other objective functions can be defined such as RatioCut (Hagen & Kahng, 1992) or the normalized cut (NCut) (Shi & Malik, 2000). Those objective functions are defined as follow:

$$RatioCut(V_1, ..., V_c) = \sum_{k=1}^{c} \frac{cut(V_k, \bar{V}_k)}{|V_k|}$$
$$Ncut(V_1, ..., V_c) = \sum_{k=1}^{c} \frac{cut(V_k, \bar{V}_k)}{vol(V_k)}$$

Where $|V_k|$ is the number of vertices in $V_k$ and $vol(V_k) = \sum_{i \in V_k} d_i$. These new objectives tend to produce balanced communities, in terms of number of edges or in terms of weights inside the communities.

It can be proved that solving these objectives is equivalent to solving optimization problems derived

from spectral decompositions of graphs (Von Luxburg et al., 2008). An approximation of the Ratio Cut can be obtained by the minimazation of the following problem:

$$\min_{H \in \mathbb{R}^{n \times c}} Tr(H^T L H) \text{ s.t. } H^T H = I \tag{4}$$

An approximation of the Normalized Cut can be obtained by minimizing the following problem:

$$\min_{U \in \mathbb{R}^{n \times c}} Tr(U^T D^{-1/2} L D^{-1/2} U) \text{ s.t. } U^T U = I \tag{5}$$

In their work, Bianchi et al. (2019) develop a pooling layer that aims at minimizing the minCUT problem. By calculating an assignment matrix $C$ as in (Ying et al., 2018) based on the projection of node embeddings on a cluster matrix, they are able to minimize problem 4 by adding a regularization term that depends on the assignation matrix $C$.

In our work, the assignment matrix $C$ does not include the information of learnable parameters since it is computed by looking at the connected components of the graph after having cut the less informative edges. But the Normalized Cut of the partition of the graph can be computed easily from the edge score matrix $S$. If we compute the matrix $M_{cut} = C^T S C \in \mathbb{R}^{c \times c}$, we obtain the super node matrix in which each diagonal parameter represents the sum of the edge weights inside each cluster (super node). Moreover, $\sum_{j=1}^{c} M_{cut_{ij}} 1_{i \neq j}$ represents the sum of the weights between cluster $j$ and the rest of the graph. With those two matrices, we can easily compute $Ncut(V_1, ..., V_c)$ at layer $l$ for a graph $G$. We thus add a regularization term equal to:

$$L_{reg} = Ncut(V_1, ..., V_c) = \sum_{i=1}^{c} \frac{\sum_{j=1}^{c} M_{cut_{ij}} 1_{i \neq j}}{M_{cut_{ii}}} \tag{6}$$

## 4 EXPERIMENTS

### 4.1 GRAPH CLASSIFICATION

**Datasets:** We choose a wide variety of benchmark datasets for graph classification to evaluate our model. The datasets can be separated in two types. 2 bioinformatics datasets: PROTEINS and D&D; and a social network dataset: COLLAB. In the bioinformatics datasets, graphs represent chemical compounds. Nodes are atoms and edges represent connections between two atoms. D&D and PROTEINS contain two classes of molecules that represent the fact the a molecule can be either active or inactive against a certain type of cancer. The aim is to classify the molecules according to their anti-cancer activity. COLLAB is composed of ego-networks. Graphs' labels are the nature of the entity from which we have generated the ego-network. More details can be found in (Yanardag & Vishwanathan, 2015).

**Experimental setup:** We perform a 10-fold cross validation split which gives 10 sets of train, validation and test data indices in the ratio 8:1:1. We use stratified sampling to ensure that the class distribution remains the same across splits. We fine tune hyperparameters $f_l$ and $d_l$ the dimensions of features in each layer, $r$ the cut ratio, $lr$ the learning rate respectively chosen from the sets $\{256, 512, 1024\}$, $\{32, 64, 128\}$, $\{10\%, 30\%, 50\%, 70\%, 90\%\}$ and $\{0.01, 0.001\}$. We do not set a maximum number of epochs but we perform early stopping to stop the training which means that we stop the training when the validation loss has not improved for 50 epochs. We report the mean accuracy and the standard deviation over the 10 folds on the test set. We compare our method with kernel methods and with graph neural networks that use pooling layers (Ying et al., 2018; Gao & Ji, 2019). We should note that kernels methods do not use node features that are available on bioinformatics datasets.

**Results:** From the results of Table 1 we can observe that our pooling layer is challenging with state-of-the-art methods. Indeed, on most datasets, the score of our model is very close to those obtained by Gao & Ji (2019) and our model outperforms Ying et al. (2018) on these datasets. From

Table 1, EdgeCut competes with all algorithms on COLLAB.

| Dataset | D&D | PROTEINS | COLLAB |
|---|---|---|---|
| Max | 5748 | 620 | 492 |
| Avg | 284.32 | 39.06 | 74.49 |
| #Graphs | 1178 | 1113 | 5000 |
| Graphlet | 74.85 | 72.91 | 64.66 |
| Shortest-Path | 78.86 | 76.43 | 59.10 |
| 1-WL | 74.02 | 73.76 | 78.61 |
| WL-OA | 79.04 | 75.26 | 80.74 |
| GraphSage (Hamilton et al., 2017) | 75.42 | 70.48 | 68.25 |
| DGCNN (Zhang et al., 2018) | 79.37 | 76.26 | 73.76 |
| DIFFPOOL (Ying et al., 2018) | 80.64 | 76.25 | 75.48 |
| g-U-Nets (Gao & Ji, 2019) | **82.43** | **77.68** | **77.56** |
| **EdgeCut** | $80.33 \pm 0.05$ | $76.42 \pm 0.23$ | $\mathbf{77.23 \pm 0.11}$ |

Table 1: Classification accuracy on bioinformatics datasets

## 4.2 NODE CLASSIFICATION

**Datasets:** For node classification we conduct our experiments on three real-world datasets, Cora, citeseer and Pubmed. They are three citations networks where nodes are articles that are linked together by an edge if there exists a citation between them. All datasets contain attributes on nodes that are extracted from the title and the abstract. Attributes represent sparse bag-of-word vectors.

| Dataset | Nodes | Features | Classes | Training | Validation | Testing | Degree |
|---|---|---|---|---|---|---|---|
| Cora | 2708 | 1433 | 7 | 140 | 500 | 1000 | 4 |
| Citeseer | 3327 | 3703 | 6 | 120 | 500 | 1000 | 5 |
| Pubmed | 19717 | 500 | 3 | 60 | 500 | 1000 | 6 |

Table 2: Statistics on node classification datasets. The classification is made with 20 nodes per class in the training set.

**Experimental setup:** We split the edges into a training, a test and a validation set according to the splits used by Kipf & Welling (2016). We fine tune hyperparameters $f_l$ and $d_l$ the dimensions of features in each layer, $r$ the cut ratio, $lr$ the learning rate respectively chosen from the sets $\{256, 512, 1024\}$, $\{32, 64, 128\}$, $\{10\%, 30\%, 50\%, 70\%, 90\%\}$ and $\{0.01, 0.001\}$. We do not set a maximum number of epochs but we perform early stopping to stop the training which means that we stop the training when the validation loss has not improved for 50 epochs. We report the mean accuracy and the standard deviation after several iterations of the algorithm on each set of hyperparameters. We compare our method with graph neural networks that uses pooling layers and with other graph neural networks referenced in table 3.
We denote by EdgeCut without regularization the version of our algorithm that isn't regularized by the minCUT term introduced in equation 6. We denote by EdgeCut the version regularized. We conduct an ablation study and we show results in table 3 to show the effects of the regularization term.

**Architecture for node classification:** In order to perform node classification with a pooling architecture we use a Graph U-Net model proposed by Gao & Ji (2019) and inspired by the works of (Ronneberger et al., 2015). Considering that images can be seen as special cases of graphs that lie on regular 2D lattices, we can have a correspondance between image segmentation and node

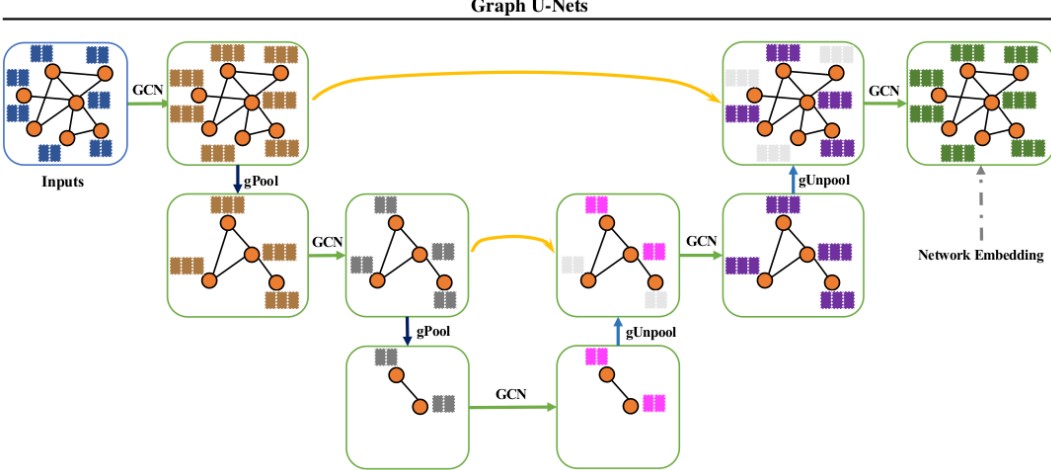

Figure 2: Graph U-Net model from (Gao & Ji, 2019). It is composed of successive convolution and pooling layers. After this encoder part, the decoder is composed of convolutions and unpooling layers. Unpooling layers reconstruct the graph and mirror the pooling layers of the encoder.

classification in graphs. By using graph convolutional layers, node information is propagated in order to identify clusters of nodes that are topologically close and that are pooled together during the training of the algorithm. During the encoder part, we store the successive pooled graphs in order to perform unpooling during the decoder part. The unpooling layer is just a mirror of the pooled graph at the corresponding level in the encoder part as illustrated in figure 2.

| Models | Cora | Citeseer | Pubmed |
|---|---|---|---|
| DeepWalk (Perozzi et al., 2014) | 67.2% | 43.2% | 65.3% |
| Planetoid (Yang et al., 2016) | 75.7% | 64.7% | 77.2% |
| Chebyshev (Defferrard et al., 2016) | 81.2% | 69.8% | 74.4% |
| GCN (Kipf & Welling, 2016) | 81.5% | 70.3% | 79.0% |
| GAT (Veličković et al., 2017) | $83.0 \pm 0.7\%$ | $72.5 \pm 0.7\%$ | $79.0 \pm 0.3\%$ |
| EdgeCut without regulazization | $81.9 \pm 0.8\%$ | $69.8 \pm 0.7\%$ | $78.7 \pm 0.3\%$ |
| EdgeCut | $82.3 \pm 0.6\%$ | $70.9 \pm 0.5\%$ | $79.1 \pm 0.4\%$ |

Table 3: Classification accuracy on node classification datasets.

## 5 CONCLUSION

In this work we developed a novel pooling layer based on edge cuts in graphs. This approach is novel because it focuses on edges to coarsen the graph. We proposed a method to compute edge scores in order to evaluate edge importance in graphs. By cutting the less informative edges, we are able to split the graph into several connected components that represent our super nodes in the coarsened level. The process of removing edges is directly linked to the problem of minCUT. By adding a regularization term that corresponds to the problem of Normalized Cut we give more consistence to the score of edges and we cut edges that allow us to split the graph into communities that are relevant to the problem of minCUT. We finally showed through extensive experiments that this novel approach competes with state-of-the-art methods.

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
