# OpenReview forum: "Graph Pooling by Edge Cut"
_ICLR.cc/2021/Conference — Reject_

### Official Review · AnonReviewer2 · 2020-10-28
**Interesting approach for the pooling layer of GNNs. Issues with the theoretical justification and empirical evaluation.**

**Rating:** 4
**Confidence:** 4

**Review:**

The paper proposes a novel pooling layer for graph neural networks. Pooling in GNNs amounts to merging nodes that are very similar through the layers. Specifically, the paper proposes to merge nodes whose edges have a high score according to the edge cuts. The edge score in practice is computed in each layer using an attention mechanism on the concatenated representations of the edge’s nodes from that layer. The authors then choose a fixed ratio r of the top edges to keep, in an effort to remove edges of nodes from distant communities. The edge score matrix which is n x n stores all the scores and is truncated based on the aforementioned threshold, renormalized, and used to extract the connected components simply from disconnected areas of the matrix. In the following layer, the clusters form super-nodes and their representations are a weighted combination of each individual node’s representation.

Strong points:

--- The overall idea is very interesting as it is utilizing the minCUT algorithm in the context of GNN, and coarsening could indeed be a promising approach to better GNN’s accuracy.

--- The paper is well-written and the different concepts are clearly presented.

Concerns:

--- There are several concerns, the main being about the novelty of the approach since the minCUT problem has been addressed already in a previous work (as the authors also mention). In this case, the authors should describe in detail how Eq. 6 and the computation of S and C result in minimizing the graph cut to underline the difference from the previous work and show that this work is not derivative.

--- The second problem adheres to the experimental section. The authors utilize very few datasets for evaluation. There is a plethora of graph classification and node semi-supervised learning datasets, ranging from OGB (https://ogb.stanford.edu/) to the Dortmunt repository (https://ls11-www.cs.tu-dortmund.de/staff/morris/graphkerneldatasets).  It is also sensible to test the model in larger networks as the coarsening may be less effective due to giant components and the distribution of edges

--- Moreover, the results do not surpass the benchmarks, even seminal ones like GAT, which calls into question the usefulness of the approach. Is there any justification for that?

--- Finally, there is a hyperparameter r which needs at least a strategy to choose it from and a sensitivity analysis to understand its effect on the final outcome.

---

### Official Review · AnonReviewer4 · 2020-10-28
**The method is novel, but the performance improvement is slight**

**Rating:** 5
**Confidence:** 3

**Review:**

Summarization

The authors propose a novel pooling layer based on edge cuts in graph, where a regularization function is introduced to produce edge scores via minimizing the minCUT problem. Through extensive experiments, the authors have proved the proposed EdgeCut pooling structure can achieve comparable performance in various graph analysis tasks.

Strong points

1) The paper is good writing and easy to understood. As far as I know, the proposed EdgeCut is a novel pooling layer based on edge cutting, which is reasonable and can explore hierarchical graph structures.
2) The authors have provided code in supplement and the experimental results are easy to follow.

Weak points

The main weakness could be the model performance and there are not enough experiments to prove the efficiency of the proposed EdgeCut.

1) For graph classification in Table.1, any thought about the reason why the proposed EdgeCut is worse than g-U-Nets.  And there is no detailed experimental analysis in Section 4.1

2) Similar question, for node classification in Table.3, the performance of EdgeCut is still worse than GAT in two of three datasets, and even only obtains a slight improvement compared to the most basic GCN in Table 3. Notably, the performance of EdgeCut without regulazization is even worse than GCN in both Citeseer and Pubmed datasets. The authors should provide more comprehensive experimental analysis and explain the reason why the performance improvement is so slight and even worse than the basic GCN.

3) There are only two quantitative experiments in this paper, additional qualitative experiments like the visualizations of graphs after pooling should also be included to prove the model efficiency, even making use of toy data.

Questions:

My questions have been included in Weak points part

Additional Feedback:

1) Can you provide time complexity comparisons for the proposed EdgeCut and other baselines

---

### Official Review · AnonReviewer1 · 2020-10-28
**Straightforward idea and weak empirical results**

**Rating:** 3
**Confidence:** 5

**Review:**

Summary:

This paper presents a graph pooling operator by first predicting scores on edges, then performing min-cut to separate subgraphs, and finally construct coarsened graphs. Authors perform experiments on several small datasets to verify their claims.

Pros:

1, The problem of defining pooling operators on graphs is important. The proposed idea is straightforward and easy to follow.

2, The writing of the paper is smooth.

Cons & Questions & Suggestions:

1, The idea is not novel. Normalized cut type of methods for graph coarsening have been investigated by quite a few works, e.g., [2]. The proposed pooling operator is not fully differentiable since the gradient through the min-cut optimization is not exploited. Therefore, it is less appealing compared to methods like diff-pool. Moreover, the empirical performance is only comparable to diff-pool on few datasets. I am further concerned about the efficiency of the proposed method since min-cut on large-scale graphs is slow which would significantly slow down the inference of GNNs. However, this part is not discussed or empirically investigated.

2, The experiment section is not that convincing. First, the performances are worse compared to other baselines. For example, the numbers in Table 1 are worse than g-U-nets consistently (I do not know why authors bold their results which are clearly not the best). Second, authors should include more graph pooling baselines, e.g., the ones with similar min-cut objectives like [2]. At last, it would be great to extensively test the proposed method on a wider range of datasets since the ones used for now are small-scale and many baselines achieve similar performances anyway.

3, I think it is necessary to discuss or at least mention the original GNN paper [1] in the literature review.

[1] Scarselli, F., Gori, M., Tsoi, A.C., Hagenbuchner, M. and Monfardini, G., 2008. The graph neural network model. IEEE Transactions on Neural Networks, 20(1), pp.61-80.

[2] Defferrard, M., Bresson, X. and Vandergheynst, P., 2016. Convolutional neural networks on graphs with fast localized spectral filtering. In NeurIPS.

Conclusion: Overall, I do not think the paper is ready for being published.

---

### Official Review · AnonReviewer3 · 2020-10-29
**Unclear explanations and limited contributions**

**Rating:** 3
**Confidence:** 5

**Review:**

This manuscript proposes a new pooling layer in Graph Neural Networks (GNN). By computing certain scores on edges which indicate the importance of edges in the process of information propagation, top r% edges are selected and a pooled graph is constructed by considering the connected components to be super nodes. The authors tried to explain some connection between their pooled graph and the normalized cut problem, which is not clearly stated in the manuscript. Even though the manuscript explores an interesting and timely topic, their approach is not technically appealing and the explanations are not enough to thoroughly understand the authors' ideas. My main concerns and major questions are as follows:

#1. In Section 3.1 (page 4), how $W_{pool}$ and $a$ can be considered as trainable variables and how these variables are actually trained are not clear.

#2. Even though the authors argue that the number of clusters needs not be specified in advance, one should determine $r$ in their method instead.  In the experiments, the authors just tried several values for this $r$ value. There should be some rules to appropriately set this $r$ value. I'm wondering if it is possible to look at the distributions of edge scores and determine an appropriate $r$ value.

#3. The descriptions about the relationship between their clustering approach (i.e., forming super nodes by taking the connected components) described in Section 3.1 and the graph normalized cut problem described in Section 3.2 are not clear. Where and how $L_{reg}$ is used in their proposed method?

#4. Experimental results do not support that the proposed method is better than existing methods. Given this limited empirical contributions, I'm wondering if the proposed method has any theoretical benefits over existing methods.

---

### Decision · Program_Chairs · 2021-01-07
**Final Decision**

**Decision:**

Reject

**Comment:**

This paper proposes a graph pooling mechanism based on adaptive edge scores that are then fed into a min-cut procedure.
Reviewers acknowledged that this is an important topic of study, but all agreed that the current manuscript does not provide enough evidence about the significance and novelty of their proposed approach.
The AC recommends rejection at this time, and encourages the authors to build from the reviewers feedback to improve upon their current work.